# Surf Therapy—Qualitative Analysis: Organization and Structure of Surf Programs and Requirements, Demands and Expectations of Personal Staff

**DOI:** 10.3390/ijerph19042299

**Published:** 2022-02-17

**Authors:** Tereza Podavkova, Martin Dolejs

**Affiliations:** Department of Psychology, Faculty of Arts, Palacký Univerzity Olomouc, 77900 Olomouc, Czech Republic; martin.dolejs@upol.cz

**Keywords:** surf therapy, organization, structure of surf programs, personal staff, qualitative research

## Abstract

Surf therapy is an increasingly popular method of mental health intervention. Although previous research documents the benefits of surf therapy for mental health, it is unclear how to navigate the organization structure of said program. This research project is a case study, where the case is an organization of our choice, in which we selected several respondents for each position. The primary study objective was to identify the principles of the surf therapy structure and describe its personnel make-up. The aim was to identify the processes in the organization, to establish their interconnectedness and outline the positives and negatives of their functioning. The respondent sample (*n* = 11) was composed of participants of a surf therapy program held in Great Britain, including staff (coordinator, surf instructor), volunteers, and clients. Data were collected through a semi-structured interview and analyzed using interpretative phenomenological analysis. Four main thematic clusters were identified (organization of surf therapy, surf therapy staff, surf session, and situations that may arise in surf therapy), as well as several other categories based on participants’ experiences. The information obtained provides a new, as yet unexplored view of surf therapy, and can be used particularly in the development of new surf therapy programs.

## 1. Introduction

Sport and exercise are known to be long-term beneficial to physical and mental health and the overall well-being of both individuals and society as a whole [1,2]. During physical activity, hormones are released, and other physiological processes occur, leading to an overall improvement in mental health [1,3,4], and improving sleep quality [5]. Regular physical activity in the natural environment can bring even greater benefits, such as reducing anger, depression [6,7] and stress [8], and contributing to wellbeing [9]. The outdoor environment offers a wide range of sensory experiences (air, temperature, sound, smell, etc.) that stimulate the whole body, although in outdoor sports, the full potential of these experiences is limited [10].

According to research, spending time in nature, especially by water, reduces the production of stress hormones, cortisol [11,12] and epinephrine [12], reducing stress [13] and anxiety [14]. The calm sound of water and waves crashing against the shore also has a calming effect [11] as well as the negative ions with which the ocean water is filled [11].

This suggests that water sports may be particularly suitable for improving mental well-being. Surfing is relatively unstructured, exploratory, and playful [15]. It is focused on experience, rather than achievement.

Adaptive surfing is formally recognized by the International Surfing Association and its mainly for people with handicaps, impairments, or disabilities, and the aim of adaptive surfing is to promote the well-being of these people [16]. An intervention that uses surfing (physical activity) with therapeutic elements of the ocean as a vehicle to achieve positive change is called surf therapy [17]. Carefully planned water activities tailored to the needs of the individual can contribute to correct psychosocial and cognitive development [18]. The International Surf Therapy Organization [19] summarizes the benefits of adequately indicated surf therapy as follows: improved physical health and mobility [18,20,21]; improved mental health [22], including reduction of specific symptoms, such as posttraumatic stress and depression [23,24,25]; improved well-being (strengthening of trust and confidence, encouragement of independence, resilience and protective coping strategies) and improved social skills [1,22]. This kind of therapy is currently provided mainly by charities and non-profit organizations regulated by the International Surf Therapy Organization, e.g., Foundation WOW in Australia, Disfrutal el Mar in Spain, Jimmy Miller Foundation in San Diego, Roxy David Foundation in Cape Town, and UrbanSurf4Kids in Portugal.

Set against the above theoretical background, our primary study objective was to explore and describe the organizational support of the surf therapy program. Our research is focused on job positions in direct contact with the client (surf instructor, project coordinator a volunteer) in surf therapy. Therefore, our three partial research targets were as follows:What are the main structures in the organization of surf therapy?What are the requirements and demands for individual positions which are in direct contact with the client?What is expected from the coordinator, surf instructor, and volunteer?

The research section applies a qualitative approach using an exploratory–descriptive technique [26].

The following research is a case study, where the case was an organization of our choice, in which we selected several respondents for each position. The aim was to identify the processes in the organization, to establish their interconnectedness, and to outline the positive and negative aspects of their functioning.

## 2. Materials and Methods

### 2.1. Participants

A purposive sampling strategy was used to recruit participants with a specific knowledge of the topic area and experiences relevant to the research questions. The final sample was selected by the snowball sampling method [26].

The sample included 11 people (6 males and 5 females; age range 20–31), who had an experience with surf therapy in some of the Wave Project’s programs. The Wave Project is a surfing intervention which has three particular characteristics: an ethos of acceptance, support and non-competitiveness [27]. They work with young, vulnerable (low confidence, poor self-esteem and self-image, lack of self-belief, social exclusion, feel stigmatized) people from 8 to 21 years in the UK and Northern Ireland [Devine-Wright and Godfrey, 2018].

We used the triangulation of data method [28], which appeared to be an effective way for providing us an insight into individual experiences during surf therapy.

Participants were divided into the three groups: 1/staff working directly with children (6 respondents), a group of participants including therapists, coordinators and surf instructors; 2/volunteers (4 respondents); 3/clients (5 respondents). Three participants had experiences with different roles in the surf therapy (for example, one boy was a client and then he became a volunteer), so we did two interviews for each position, separately.

All participants provided oral informed consent for the study and also for the recording of the interviews. The data was anonymized and used only for research purposes.

### 2.2. Methods

First, we used an analysis of secondary official documents which included an Annual Report from 2019 [29], an independent longitudinal evaluation of the impact of The Wave Project on vulnerable young people 2013–2017 [30], an independent evaluation of The Wave Project’s impact on vulnerable young people over 3 years from 2013–2015 [27], and a piece of independent research [22]. This part of the research process was used to complete the information obtained from the interviews.

Furthermore, an interview plan was created for each group. The plans focused on personal experience with surf therapy, specifically in three areas: 1/organization of surf therapy, 2/staffing of surf therapy, and 3/surf sessions.

### 2.3. Pilot Study

Following the development of questions for the semi-structured interview of all groups, and prior to the research itself, a pilot study was conducted to check the respondent understanding of the questions. This involved a small number of respondents, specifically five persons actively involved in surfing, who were not part of the research discussed in this paper. Based on the survey, the wording of selected questions was changed. For example, the question “Are there any contraindications for this type of therapy?” was reworded to “Are there any reasons why the client cannot enter the program/or is terminated during the program?” No question was removed or added.

In the next part of the pilot study, an interview was conducted with one respondent from each group, where we focused on understanding the questions, and especially on adding or removing some items, which did not happen.

### 2.4. Data Collection

Interviews were conducted between October 2020 and February 2021, with a mean interview time of 60 min for the staff and 30 min for volunteers and clients.

Online video conferencing was used for 5 interviews for logistical reasons. The remaining 13 interviews were conducted in person at locations suggested by participants including local cafes, participant’s homes, or on the beach.

Interviews were conducted through information technology. The time and technology of audio and video calls were adapted to the possibilities and willingness of individual respondents.

### 2.5. Data Analysis

We used interpretative phenomenological analysis [31]. We followed the general analytical procedure of interpretive phenomenological analysis, as recommended by Smith, Flowers, and Larkin [32]. Initially, we read the text repeatedly and, if necessary, we listened to the recording again. In the case of capturing important information, we supplemented the text with notes on the left side of the text. After processing the last interview, we started to organize the obtained data and looked for emerging topics, which we recorded on the right side. All topics were written into one list. Based on the similarities of individual topics, we created thematic clusters, from which sections were published in the end.

## 3. Results

Figure 1 illustrates the main clusters and their categories. The clusters gave rise to several other categories to enable a better insight into the issue under study. The main clusters and their categories are discussed in detail in the text below the image. For anonymity, we used pseudonyms instead of participants’ names.

### 3.1. Core Categories

#### 3.1.1. Specifics of Surf Session in Surf Therapy

In regular surfing sessions, children are mostly enthusiastic about surfing: children have usually persuaded their parents to try surfing. However, in contrast, at surf therapy sessions, children have reduced self-esteem and a tendency to give up on first failure. One of the surf instructors added: “*I feel that I had more physically fit children in a ordinary surf session.*” Furthermore, it is necessary to motivate these clients more, and talk a lot with them.

Sometimes it can happen that the client will not surf, instead he will jump in the water and talk to the volunteer: “*Sometimes we didn’t surf much but it was about building sandcastles… then we went into the water, and the girl just sat on the surfboard and talked or jumped into the water*”.

#### 3.1.2. Organization of Surf Therapy

##### Conditions for Inclusion in the Surf Therapy Program

The only age restriction was 8 to 21 years. Exceptions were possible under specific circumstances and needed to be adequately justified.

Clients enter the surf therapy program based on a referral from a registered referral partner, such as a doctor, school counselor, social worker or psychologist [The Wave Project, 2020]. The referral is an online form. Parental agreement is required as the form asks the applicant to complete the contact details of the parents (or foster parents or children’s home). Likewise, the diagnosis and the reason why the person is referred need to be completed. Referrals from different professionals varied in extent and length. One of the respondents said: *“Usually the doctor wrote three sentences, but the recommendation from a psychologist or social worker had three paragraphs.”*

##### Admission of the Client to the Surf Therapy

Clients are admitted by phone. The coordinator contacts the legal representative to determine if the family is still interested in surf therapy and informs them about the waiting period and the nearest opening.

A welcome letter, a questionnaire for parents and children, and a small gift are sent. The questionnaire includes questions such as “How are you feeling?” and “What will make you happy?” The same questionnaire is completed again after a six-week interval, to compare the therapy effect on the person concerned.

The document needs to be signed by the legal representative and includes informed consent for data processing, consent to the provision of the therapy to the child, permission to photograph and record the young person (GDPR), and permission to drive the young person to/from surfing sessions (if interested).

#### 3.1.3. Surf Therapy Staff

A large number of people are involved in the organization and course of surf therapy. The whole process begins in the office, where administrative staff, organizers, researchers and, given the charitable aspect of the therapy, fundraisers work. The following positions are in direct contact with clients during the therapy.

##### Coordinator

The principal requirement for joining the program is a DBS check (Disclosure and Barring Service). Coordinators need to know basic facts about the ocean, have water skills and at least basic surfing experience. However, the coordinator does not need to be a surf instructor.

The coordinator is in charge of surf therapy programs and their organization. They analyze all the recommendations from professionals, on the basis of which they communicate with parents, guardians or carers. They evaluate clients, based on the information obtained, as: 1/high risk client, 2/moderate risk client, and 3/low risk client. The classification depends on diagnosis, health, and ability to swim. All of this determines the number of people the client needs.

Last but not least, the coordinator recruits instructors and volunteers, and does so by effectively promoting the organization, which is another responsibility of the coordinator. The coordinator is also in charge of further training of those involved in the therapeutic programs.

##### Surf Instructor

One respondent said that “*… an instructor has to be able to give clear and simple instructions, to demonstrate how to surf to make it fun and a truly positive experience.*”

Again, DBS check is the main requirement for this position, as well as a surf instructor license and an open water safety qualification. Respondents pointed to the variety of licenses available and their quality. They specifically praised the course organized by the International Surfing Association, which is the international standard for the accreditation of surfing coaches and instructors.

Given the specific client group, the instructors agreed on patience as the decisive factor.

Although respondents agreed that knowledge of psychology or psychotherapy is useful, psychological training is not a job entry requirement. It is therefore up to the instructor whether they become more familiar with the issues of the young person they are in charge of. Peter says: “*I was aware of their diagnoses, so I’d try to read about it and prepare what I should do and what I shouldn’t do.*”

Two respondents stated that while they were working for The Wave Project, the charity offered optional selected crisis training. The training was organized by the coordinator.

##### Volunteer

An important part of the project is the volunteers, who have a number of key roles in water and on land. Their determination and care contribute to the improvements observed in young people who have completed the surf program [29]. Former surf therapy clients or their family members may also become volunteers, and thus help other people build confidence through surfing [30].

Anyone wishing to join the project as a volunteer needs, as with previous positions, to have a DBS check. Concerning volunteering, Caroline said that “*a good volunteer should definitely have a lot of patience and be positive … a positive attitude and enthusiasm, and the ability to make the clients feel the same.*”

The main idea was to pair each client with one volunteer. In the case of a lack of volunteers, one volunteer may be in charge of two clients. This may happen in Surf Club, while none of the respondents reported lack of volunteers in the initial course.

Coordinators also strive to ensure relationship continuity. Sessions are typically run by the same volunteer, and each course has its permanent support staff.

A training session is held for the volunteers several times a year. The session is designed for volunteers and new volunteer applicants. It is about two hours long and the first part is devoted to the description of surf therapy and volunteer requirements. The other part of the training session is dedicated to practice and skill training. Participants send each other out to the water to experience what it is like to completely rely on and trust another person. This gives them the client experience. Joseph described his experience with the training session: *… for example at that first practice, I had my person ride out the first wave to a total nosedive (the surfer falls of the board by pitching the nose down).*”

Respondents agreed that the greatest challenge for a volunteer was the responsibility and “*the ability to handle, from time to time, a tense situation that may take place.*” They may consult instructors or coordinators about challenging situations, who will help them.

##### Supervision/Expert Session

It is obvious from all the interviews that there was no supervision organized for coordinators, instructors, or volunteers. However, they met voluntarily, and they often returned to individual cases together.

#### 3.1.4. Surf Session

##### Transportation

The Wave Project works with another NGO to provide the transportation of children to sessions. The option is available to disadvantaged families who cannot afford to drive their children to surf sessions.

##### Surf Session

The team of volunteers and instructors, headed by the coordinator, arrive at the beach about a half hour before the session to prepare the equipment.

Clients arrive by themselves or use the transportation option (see above). They are told to arrive 15 to 20 min before the session to be ready on time. After arriving at the beach, the clients are grouped with instructors, who, together with volunteers, help the children put on the wetsuits. As Catherine said: “*changing into the wetsuit is an icebreaker, the kids tend to befriend someone and want to stay with them. So, we just wait and see what happens, if they want and find someone.*”

The program opens on the beach with a warm-up, ideally a beach game. This is followed by a quick explanation of safeguarding policy and brief information about surfing. The clients try stand-up (pop-up, take-off) and are shown how to ride a surfboard. They first try the technique lying down, then on their knees and, if interested, they try standing up (“*… we don’t want them to have a perfect posture, it’s more about just standing up… that’s not our goal number one.*”).

Every session is run by two surf instructors. Joseph described their roles: “… one instructor is in the shallow water facing the ocean, watching, and the other is with the kids” The sessions are one-on-one, if possible, with each young person paired with one volunteer, although the number of volunteers can be lower in exceptional cases. In this case, surf instruction is adjusted accordingly (for example, all volunteers and instructors remain with the kids in waist- or knee-deep water), as Caroline said: *“We make sure it is safe.”* The client decides how deep they want to go. They may stay in knee-deep water if they do not feel like going any further.

Harry said that “*… the most important thing is to listen what the kids want… feel sure they are safe… and work only as fast as the kid so that they have fun.*”

The time spent in the water ends about twenty minutes before the end of the session. Once everybody is back on the beach, the session is briefly summarized and evaluated. The volunteers help clients take off the wetsuit and hand the clients over to parents/carers or drivers.

##### Surf Therapy Kit

Foam boards and bodyboards (shorter than surfboards) are used. The Wave Project uses special equipment, namely adapted surf boards, to allow people of all abilities to surf [The Wave Project, 2019]. The boards resemble the boards used by lifeguards, which have four handles on the side to enable clients to hold on to them.

The surf equipment and wetsuit are lent by The Wave Project. All the clients need to bring for the session is a towel or poncho.

#### 3.1.5. Situations That May Arise in Surf Therapy

##### Adverse Weather Conditions

Surf conditions in Great Britain are not always suitable (e.g., large waves, onshore). The program always has a backup plan ready, which features beach games or bodyboards. Isabelle says that “*… if the tide doesn’t work out, we know… and in England, the difference is huge, six metres easily. And so, when we could not very well adapt the session to the conditions, we’ll adapt the content. If there are no waves and it’s totally flat, we for example stack the surf boards one next to another in the water, making a sort of a bridge and have the kids run there and back… and we just invent other things to do.*”

##### Uncooperating Client

A child may not wish to participate in a particular surf session. Such situations are dealt with on an individual basis. The clients are not pressured at all in the initial course. As Joseph noted, once “*the kid just sat on the beach throughout the whole session. I once had a kid sitting on the beach for 3 weeks in a row, just playing with sand, and it was not until week four that the kid went in the water… and it took until week six for the kid to get in the water up to its neck.*” Clients are encouraged to participate at a greater extent in the follow-up Surf Club, mainly due to the lower number of volunteers at the sessions.

##### Rejection of a Specific Volunteer, Instructor

The majority of respondents could not remember a single occasion when a child would specifically refuse to work with a selected volunteer or instructor. Coordinators always strive to have enough instructors and volunteers on the course to ensure the client does not need to meet new staff every surf session.

One of the respondents, however, described a situation they experienced: “*… I must have tried to motivate the kid too hard and the kid was afraid… and then a wave came over us and the kid got scared and refused to be in the water or around me. But it turned out temporary… I began to play with the kid and soon we were best buddies again and we surfed until the end of the session.*”

##### Attachment to a Particular Volunteer, Instructor

Bonding with a volunteer or instructor is more common than rejecting one in surf therapy. The bonding is encouraged, and coordinators work towards the continuity of the relationship. Sometimes, a young person may work with the same volunteer for the whole of the six weeks. Friendship is thus nourished, not only among participants in the surf therapy program, but also among families. This helps build a community.

If a young person becomes attached to an instructor in charge of a paralyzed (high risk) child, the instructor explains the situation to the person. Joseph commented: *“If there is another, completely paralysed child, whom I cannot transfer to a volunteer, because you just don’t, as I run the course and I have first aid qualifications and have worked with this… so I explain it to them, say like we’ll catch three or four waves at the end but for now I have to be with the child because they need me.”* None of the respondents remembered an occasion where the client was not understanding in such a situation.

##### Conflicts in Session

It may happen that clients do not get along during the session. The situation may escalate, and conflict needs to be addressed. “*Well, I remember that once a girl was riding the wave standing, a boy grabbed some sand, thought it was fun and threw the sand next to her, but she fell over and blamed him… although it had nothing to do with him. And so, she became cross, aggressive.*” These events are common, according to the respondents, and easy to get under control.

Conflicts are much more common in Surf Club, where the clients know one another. Catherine says that “*once in Surf Club we spent an hour out of water, discussing the rules, that was pretty common. One of the boys got a month-long time out of Surf Club as he had physically bullied a kid*.”

##### Opening a Sensitive Topic in Session

The trust the client builds during the program toward a volunteer may prompt the client to confide in the volunteer. Any such situation has a clear outcome. The volunteer may listen but never comment, nor request details. The volunteer is obliged to report the information to the instructor in charge of the session. A protocol needs to be completed (the protocol is also completed in the case of injury) and handed over to the coordinator, who takes over the case.

The next step depends on how serious the reported information is. The majority of cases involved a family problem that required an approach not involving the parents/carers (“*Sometimes it was about the parents, psychological abuse, neglect.*”). The coordinator gathered information and reported it to social services.

#### 3.1.6. Responses to Research Questions

Detailed planning is critical at the onset of preparations of a surf therapy program. The plan needs to define the target client group and ways of implementing changes in physical and psychosocial health (e.g., surfing, psychoeducation).

What Are the Main Structures in the Organization of Surf Therapy?
Referral to the program (from the physician, psychologist, social worker) and current health status of the individual;Informed consent completed by client (signed by parent/carer in the case of minor clients);Therapy effect questionnaire developed by the organization, before and after each surf therapy course;1:1 volunteer–client and relationship continuity (volunteers do not alternate within the program);Volunteers may include former surf therapy clients or family members;Organization equipment;Transportation option;Practice session.What Are the Requirements and Demands for Individual Positions which Are in Direct Contact with the Client?

Coordinator:DBS check (Disclosure and Barring Service);Must have basic knowledge about the ocean, water skills, and at least basic surfing experience; does not need to be a surf instructor;Point of contact for clients and their parents/carers, ensures all their needs are being met.Surf Instructor:DBS check (Disclosure and Barring Service);ISA Level 1 Surf Instructor qualification;Beach lifeguard qualification;Must be friendly and enthusiastic, flexible, punctual and organized.Volunteer:DBS check (Disclosure and Barring Service);Safeguarding training;Must be friendly and enthusiastic, patient and responsible.

3.What is Expected from Coordinator, Surf Instructor and Volunteer?

Coordinator:Evaluates risk for each client based on the referral;Works alongside surf instructors (before, during, and after surf session and provides them with information about clients);Organizes session volunteers and provides them with training prior to the course;Attends surf sessions and makes sure all safeguarding measures are complied with;Promotes the development of a community in the project location;Keeps complete and precise records about all referral organizations;Actively raises funds to ensure the project continuity in the location (funding requests, community fundraising, and contributions from referral partners) [33];Promotes organizations, updates the local page of the website and social media.Surf Instructor:Organizes group weekend sessions for vulnerable children and young persons aged 8–21 years with different level of other needs;Must adapt the surf session to meet the client needs.Volunteer:Mentors and supports clients during surf sessions.

## 4. Discussion

This research project describes a new therapeutic intervention, which the preceding research [10,17,18,23,34,35,36] has shown to have a positive effect on mental health. Only a few research studies have studied the topic, largely due to the novelty of the topic.

Surf therapy participants tend to have prevalently positive experiences with the intervention. The experience is also often described in the literature. In our research, respondents and clients reported improvements in mental health and the interpersonal area. The following benefits in the social area have been described in preceding research studies [1,37]: social integration, community, and contact with other people with difficulties. By learning a new skill as part of a social group, the surf therapy clients developed important skills, such as social trust, resilience, empathy, responsibility, making new friends, and improved communication strategies [18,35]. Surf therapy has a positive impact on clients as well as volunteers and surf instructors [30]. A great advantage of surf therapy is its broad range of application, e.g., it is appropriate for children with autism [38,39], posttraumatic stress disorder [23,24,36], depression and anxiety [24,25], children from disadvantaged families [37,40], physical disability [10], etc. Other reported benefits include improved self-confidence and resilience in the majority of respondents.

We found only two case studies on surf therapy [41,42]. The benefits surf therapy has for individual areas are not discussed in the research studies, chiefly because the sessions are attended by multiple people with different difficulties. Further research could therefore focus on, for example, specifying the benefits of surf therapy for the individual areas of application in the sense of where exactly and what in particular helps. Case studies render the best insight into a new topic, and provide new inspiration for large-scale quantitative studies.

Current qualitative and quantitative data focuses primarily on how surf therapy affects clients’ lives, which is understandable given the novelty of the topic. From our point of view, the key is who provides the surf therapy. Therefore, within the staffing of surf therapy, we consider it very important to correctly define the roles (their expertise, competencies, etc.) that come into direct contact with the client and to define their sphere of work.

Drawing on the publicly available information and the results of our research project, we created an overview of the requirements for each position (coordinator, surf instructor, and volunteer), which is shown in Figure 2. Our data indicated the absence of any requirement for psychological training. If surf therapy is to have the maximum possible benefit for clients, we propose psychological training be required for at least the position of coordinator, who needs to train instructors and volunteers in specific psychological areas (e.g., instructors working with an autistic child need to have basic knowledge of these disorders and be aware of how to work with such a child). This requirement is vital, as the therapy serves as a mental health intervention.

Positions directly working with clients need to, in our opinion, have access to regular supervisions.

Approximately ten percent of people aged 14 and older become voluntary help in the The Wave Project [22,30]. This role overlap may be highly beneficial. The positive impact of volunteering is mainly in the extra benefits resulting from the perspective of the individual as a client. Surf therapy benefits not only clients but also instructors and volunteers. One of the respondents remained in surf therapy as a volunteer, enjoying the advantages of the therapy itself. He said: “*… I am now much more confident… for example when working with people.*”

Another factor is the lack of work with the family. Disability may affect not only the individual themselves but the whole family. The Wave Project allows the family to become involved as volunteers. One thing worth considering is engaging the family in training and therapy. Psychoeducation of both the family and the client could be advantageous for all the parties involved. Understanding one’s problems and knowledge of coping strategies may effectively influence quality of life. We therefore deem psychological training essential at least for one position directly working with clients.

Our findings showed that surf therapy does not include any targeted psychotherapeutic work. This was also highlighted by one of the respondents: “*When you are in the water, there’s lots of moments you need to respond to… often you are in a situation where you need to be resilient… if the people had some therapeutical background, there could be some kind of reverberations and you could work with it outside the water… that would work more.*” The use of a specific psychotherapeutic approach could boost the surf therapy effect. The integration of psychotherapy into surf therapy and the study of its impact on the client’s condition could thus constitute another topic for future research.

The question of whether this type of therapy could be applicable inland concludes the discussion chapter, and serves as yet another suggestion for further research. Based on available information, the principal mediators of surf therapy are community, exercise, and the natural environment (the positive effect of the ocean was described in terms of surf therapy). These are shown in Figure 3.

Inland areas would lack one of the elements mentioned above, namely, the ocean. Studies [6,9,10] define physical activity in the natural setting as beneficial for mental health. Both physical activity (e.g., wake surfing, wakeboarding, river surfing) and community are preserved in our proposal. This part is, in our view, a topic for further research.

Our literature review did not yield any research project dealing with the inland implementation of surf therapy. A pilot study into out-of-ocean surfing is currently under way in Great Britain. The respondent group comprises 30 elementary school children with mental health difficulties. The six-week-long program features a weekly two-hour session with a surf instructor in a surfing wave pool. If this proves to be an effective surf therapy option, surf therapy will be expanded by inland surfing [43].

## 5. Limitation

A large number of organizations currently provide surf therapy programs under the International Surf Therapy Organization (ISTO), which strives to set general guidelines for the intervention. Yet, the organizations differ in their concepts of surf therapy, as confirmed by our first respondent trained in psychology and psychotherapy. She said: “*Each organization is different. I know several organizations doing what they call surf therapy, but there are huge differences.*”

Our research work is based on a single organization—The Wave Project. This limited us to interpreting the data obtained only within this institution, and the findings thus cannot be extrapolated to the entire population or other organizations. Conducting this study with programs within the larger ISTO organization would provide more useful and generalizable data.

## 6. Conclusions

Eleven respondents allowed us to view surf therapy from their perspective in our research work. Interviews with the participants provided important insights into aspects that were missing in the literature. We studied in detail the organization and staffing of surf therapy and we identified situations that are likely to occur in surf therapy (e.g., adverse surf conditions, conflicts during surfing).

Our findings inspired us to discuss the possible use of a new therapy option in inland regions, and to innovate and improve the present surf therapy.

## Figures and Tables

**Figure 1 ijerph-19-02299-f001:**
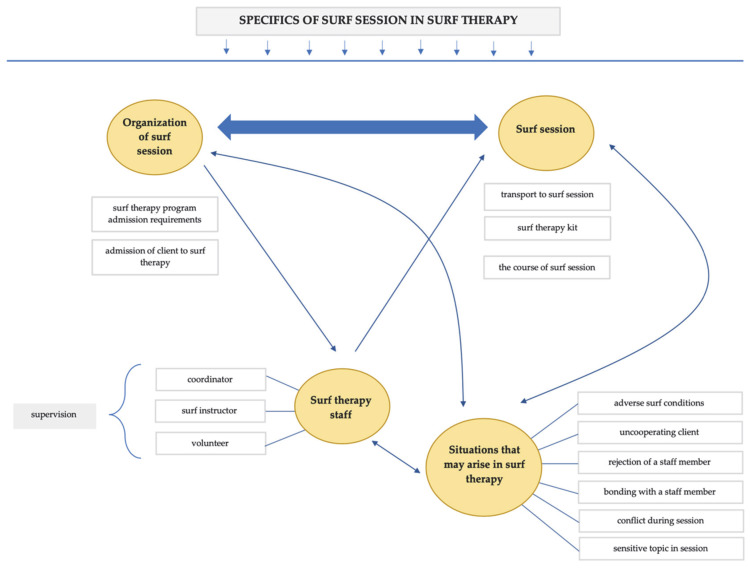
Main clusters and their categories.

**Figure 2 ijerph-19-02299-f002:**
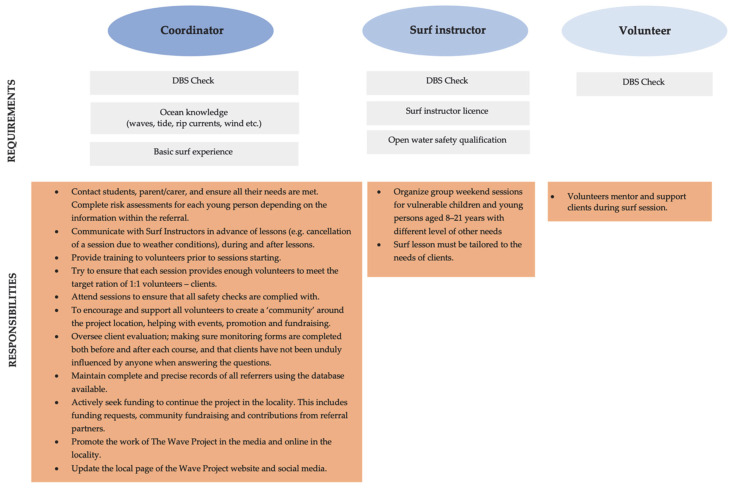
Surf therapy staff—positions directly working with clients.

**Figure 3 ijerph-19-02299-f003:**
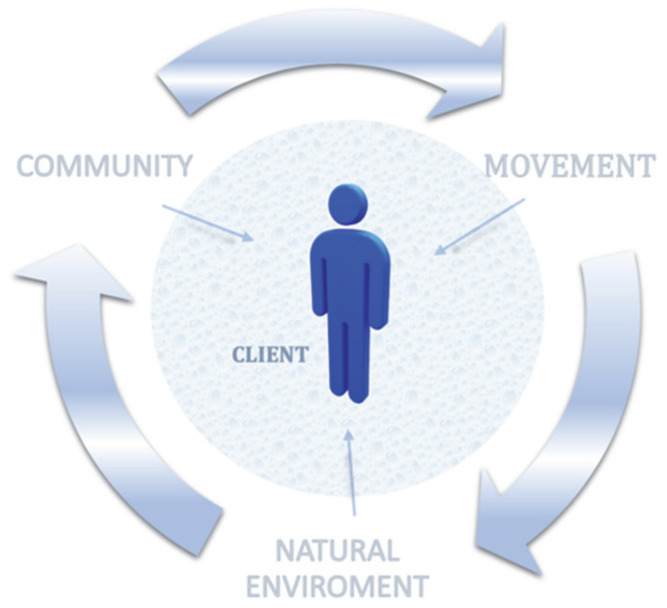
Mediators of the surf therapy process.

## Data Availability

The data presented in this study are available on request from the corresponding author: tereza.podavkova@gmail.com.

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
