# Peer review of "Surf Therapy—Qualitative Analysis: Organization and Structure of Surf Programs and Requirements, Demands and Expectations of Personal Staff"

_ijerph, 2022, doi:10.3390/ijerph19042299_

Round 1

Reviewer 1 Report

The introduction and methods sections could have a bit more detail in them.

79: each not ech

267: delete of

454: natura should be natural?

Reviewer 2 Report

The current manuscript presents semi-structured interview data from 11 individuals involved with the Wave Project, including staff, volunteers, and clients. The authors describe 4 aspects (themes) of the program that were derived from these interviews, including the organization of surf sessions, surf therapy staff, the surf session itself, and situational factors. Furthermore, the authors provide ideas for requirements for surf therapy program staff.

There are several main concerns that emerged during the review of the manuscript, which will be described below:

  1. A central concern is the sample, which the authors acknowledge as a limitation. The sample includes 11 individuals recruited by snowball sampling within the same program. The generalizability of these 11 individuals to the program is questionable, let alone to other surf therapy programs. The authors even include a quote about the diversity of surf therapy programs. Conducting this study with programs within the larger ISTO organization would provide more useful and generalizable data than a select group of individuals within one organization.

  1. Another main concern is the focus and utility of the manuscript. There appears to be two areas presented, including (1) the 4 aspects of the program and (2) defining roles and responsibilities for the surf therapy program staff. The manuscript would be strengthened if it were focused on one of these areas (the staffing content may have greater impact given the data), and all the content were directly related. As the manuscript is currently written, it is difficult to determine what information is most important to take away and how it can be used. In a similar vein, the authors could consider writing an article about the ethical considerations for surf therapy staff. The authors mention on several occasions that psychological knowledge may not be required for instructors and other staff; however, lack of this knowledge could potentially present ethical dilemmas. A manuscript that focused on this information could offer important insight for those involved with surf therapy programs. Similarly, the situations that arise section could be expanded upon and presented in a brief report. In sum, the manuscript does not appear to have a central thesis and would benefit from a more focused presentation of the data.

  1. The literature included in the manuscript needs to be checked for accuracy and overall, would benefit from restructuring to support the main research questions more clearly. There were citations that did not correctly reference the research articles that support the given finding. Several corrections to citations will be provided below.

Minor concerns:

  1. Is a year missing from the Wave Project cite in line 30?

  1. In lines 30-31, should the sentence read “It is focused on experience, rather than achievement?” (i.e., removing the “not” from the first clause).

  1. The ISTO 2018 citation should not be the sole citation for these findings, and rather citations should be provided for each outcome mentioned.

  1. Lines 40-42 read as abrupt, and it is unclear how they are related to the larger study. Please see point #2 above for further comments about the focus of the manuscript.

  1. Please define “vulnerable” in line 61.

  1. The authors state that there are 11 participants in the sample; however, they indicate there are 6, 4, and 5 respondents in each of the 3 groups (lines 65-67). Please correct or clarify.

  1. Were study participants consented for this research? The authors mention consent for surf therapy participation, but it was not entirely clear regarding research participation.

  1. In the situations that arise, how were the subcategories defined? It appears some could be more generally written. Also, there were none that mentioned other participants – was this an oversight? Or simply not mentioned by study participants?

  1. In line 124 – it is unclear what the quote is referring to in terms of a “regular surf session.”

  1. The first names can be removed from the text.

  1. The works by Caddick and colleagues could be cited in line 389 pertaining to social impacts of surf therapy.

  1. In lines 395-396, Otis and colleagues (2020) could be cited in reference to both PTSD and Depression.

  1. In lines 396-297, Glassman and colleagues would not be the best citation as this study evaluated gender differences in surf therapy outcomes. The parent study from which these data were derived is a more accurate citation, which is Walter and colleagues (2019).

  1. The mention of case studies in line 403 does not seem relevant to this research and could be removed.

Reviewer 3 Report

This research topic is very interesting, and it is grateful that the authors decided to improve the research on this field. It is of a great importance to use qualitative research methods on the field of adaptive sports and to be aware of a more subjective perspective. Also, the authors developed the research on a less explored theme, such as, surf therapy.

However, I consider that you should perform several changes in the manuscript:

Introduction

  • to perform a better explanation/ review and necessary citations of the benefits of physical activity and water sports
  • to briefly explain the advantages of surfing to approach the idea of line 28 and 29
  • to explain what the wave project is and define what is surf therapy and adaptive surfing
  • To indicate other examples of surf projects (other countries), regulated by International Surf Therapy organization
  • The information present in the lines 40 and 41 might be rearranged in de description of what is the wave project/ surf therapy

methodology

  • The ethical concerns may be better explained. It is not specified how the participants gave their informed consent for the interview and for recording activities.

Discussion

  • the authors must be careful from lines 394 to 400, because they are considering surf therapy in general and do not differ from adaptive surf classes
  • In lines 458 to 459 the authors must consider be careful with the citations, more citations and more recent

References

  • seven references in 30 (24%), have 10 or more years; the authors must be careful with citation to strengthen the research topic.

Round 2

Reviewer 2 Report

%MCEPASTEBIN%

The revised version of the manuscript is improved after addressing reviewer concerns. However, there are additional changes that could be made to further strengthen the manuscript:

  • The author’s response indicated that this is a case study of the personnel involved with the Wave Project. This conceptualization seems to be a reasonable way to frame the study; however, it is not explicitly stated in the text. It would be helpful if the authors included mention of the case study approach in at least the Abstract and Introduction.  
  • Although the authors integrated some of the recommended references, there are still issues with appropriate citations throughout the manuscript. Here are some noted examples: 
  • In line 53, the only mental health citation that is referenced is Godfrey et al., 2015 and there are no citations for the latter part of the statement about specific symptoms despite Rogers et al., 2014; Otis et al., 2020; and Walter et al., 2019 examining specific symptoms, such as posttraumatic stress and depression.
  • Related, the authors cite Walter et al., 2019, but it is a methodology article and not the article evaluating surf therapy outcomes (also published in 2019, but in the Psychology of Sport and Exercise).
  • There is a Scoping Review of surf therapy published in a Special Issue of the Global Journal of Community Psychology Practice (Benninger et al., 2020) that provides an excellent synthesis of the literature. Adding related content could strengthen the Introduction and provide the authors with a solid overview of the published literature (including other relevant citations to incorporate). 
  • The Introduction could benefit from reorganization. There are several separate statements about surf therapy that could be integrated into 1-2 paragraphs to more cohesively present what surf therapy is, why it could be beneficial, and what are associated outcomes.

Author Response

Thank you for your suggestions. We reviewed the whole manuscript and edited it as recommended.